# Gene Editing of the Follicle-Stimulating Hormone Gene to Sterilize Channel Catfish, *Ictalurus punctatus*, Using a Modified Transcription Activator-like Effector Nuclease Technology with Electroporation

**DOI:** 10.3390/biology12030392

**Published:** 2023-03-01

**Authors:** Guyu Qin, Zhenkui Qin, Cuiyu Lu, Zhi Ye, Ahmed Elaswad, Yulin Jin, Mohd Golam Quader Khan, Baofeng Su, Rex A. Dunham

**Affiliations:** School of Fisheries, Aquaculture and Aquatic Sciences, Auburn University, Auburn, AL 36849, USA

**Keywords:** channel catfish, transcription activator-like effector nucleases, follicle-stimulating hormone, hormone therapy

## Abstract

**Simple Summary:**

Follicle-stimulating hormone (*fsh*) is crucial to the reproduction of fish. In the current investigation, transcription activator-like effector nuclease (TALEN) plasmids targeting the channel catfish *fsh* gene were electroporated into fertilized eggs to generate infertile channel catfish (*Ictalurus punctatus*). Human chorionic gonadotropin (HCG) hormone treatment enhanced the spawning and hatching rates of catfish *fsh*-knockout mutants when lowered infertility was detected. Gene editing of channel catfish offers promise for reproductive confinement of genetically modified, native, and invasive fish to eliminate genetic drift into external ecosystems.

**Abstract:**

Follicle-stimulating hormone (*fsh*) plays an important role in sexual maturation in catfish. Knocking out the *fsh* gene in the fish zygote should suppress the reproduction of channel catfish (*Ictalurus punctatus*). In this study, transcription activator-like effector nuclease (TALEN) plasmids targeting the *fsh* gene were electroporated into fertilized eggs with the standard double electroporation technique. Targeted *fsh* cleavage efficiency was 63.2% in P_1_
*fsh*-knockout catfish. Ten of fifteen (66.7%) control pairs spawned, and their eggs had 32.3–74.3% average hatch rates in 2016 and 2017. Without hormone therapy, the spawning rates of P_1_ mutants ranged from 33.3 to 40.0%, with an average egg hatching rate of 0.75%. After confirmation of the low fertility of P_1_ mutants in 2016, human chorionic gonadotropin (HCG) hormone therapy improved the spawning rates by 80% for female mutants and 88.9% for male mutants, and the mean hatch rate was 35.0% for F_1_ embryos, similar to that of the controls (*p* > 0.05). Polymerase chain reaction (PCR) identification showed no potential TALEN plasmid integration into the P_1_ channel catfish genome. Neither the P_1_ nor the F_1_ mutant fish showed any noticeable changes in in body weight, survival rate, and hatching rate when the reproductive gene was knocked out. F_1_ families had a mean inheritance rate of 50.3%. The results brought us one step closer to allowing implementation of certain genetic techniques to aquaculture and fisheries management, while essentially eliminating the potential environment risk posed by transgenic, hybrid, and exotic fish as well as domestic fish.

## 1. Introduction

The genetic modification of catfish has the great potential to increase yield, disease resistance, environmental adaptation, and the release of particular proteins [1,2,3]. Nevertheless, the possible genetic and ecological implications of these fish have raised some concerns [4]. Effective fish sterilization technology could eliminate or significantly lower the environmental threats posed by transgenic fish.

Gonadotropins (GTHs), such as follicle-stimulating hormone (*fsh*) and luteinizing hormone (*lh*), have been discovered to regulate the in vitro production and discharge of steroid hormones to drive ovarian maturity in female fish and spermatogeny in male fish, respectively [5,6,7,8,9]. The gonadotropin *fsh* is a member of the glycoprotein hormone family [10]. Functional glycoproteins consist of a complex of α and β subunits. All α subunits are identical within a single species of fish; however, the different β subunits communicate the hormone’s physiological specialness [11]. In salmon, the maximal levels of *fsh* formation and release correspond to the yolk formation stage (vitellogenesis) in the annual breeding cycle, whereas *lh* generation peaks amid egg maturity and spawning [12]. Salmonid fish have high *fsh* levels during spermatogenesis and optimum *lh* levels in the spawning season according to [13]. During the breeding cycle, the occurrence of oocytes at different stages of maturation necessitates the simultaneous presence of ovarian *lh* and *fsh* levels [14]. The periodicity of hypothalamic activity level and the characteristics of gonadotropin receptor activation in the gonads provide evidence of a fish’s proper reproductive system [9]. Insulin-like growth factor 1 (Igf1) and growth hormone, in addition to gonadotropins, influence the ovaries’ maturation process. Igf1 promotes the rate of oocyte maturation and increases the production of essential sex steroids for maturation [15]. Our hypothesis is that *fsh* gene mutations cause abnormal gene functions and finally sterilize the fish.

This study contains the goals as follows, (1) utilize transcription activator-like effector nucleases (TALENs) and double electroporation technologies to knock out the *fsh* gene, (2) produce genetically engineered sterile channel catfish (*Ictalurus punctatus*) via TALEN-mediated mutagenesis at the DNA level, (3) evaluate the reproductive ability of mutant catfish via fish mating experiments, (4) examine F_1_ fish for mutations to confirm genetic inheritance, (5) bring back the fertility of *fsh*-mutant fish through injecting hormones, and (6) assess the pleiotropic effects of gene knockout in the F_1_ offspring. The effective sterilization technology could be used to ensure biosafety and prevent catfish and other gene-edited fish from spreading out in the environment. This technology could also be applied to domestic and invasive fishes.

## 2. Materials and Methods

### 2.1. TALEN Plasmid Construction and Preparation

TALEN plasmids were constructed by Transposagen Company (Lexington, KY, USA). The schematic representation of TALEN plasmids is showed in Appendix A. The β subunit of the channel catfish *fsh* gene was chosen for the TALENs’ target sites. The *fsh* gene (GenBank Accession number is NM_001200079.1) confers the physiological specificity of the hormone. The left DNA binding site is 5′-TACCAACATCTCCATCAC, and the right DNA binding site is 5′-TGTGGCAGCTGCATCA-3′. TALEN plasmids were replicated, extracted, and diluted followed the protocol described in Qin et al. 2022 [16].

### 2.2. Broodstock and Gamete Preparation

The Kansas Random strain of channel catfish were chosen as the experimental brood stock. The luteinizing hormone releasing hormone analog (LHRHa) (Reproboost^®^ Implants, Center of Marine Biotechnology, Columbus Center, Baltimore, MD, USA) with a dose of 90 μg/kg was implanted into the fish for spawning. Fish eggs were hand stripped and 200 eggs were collected for each group. Double electroporation was performed to transfer the TALEN plasmids into the eggs [17]. Our lab has applied double electroporation for plasmid delivery to make transgenics for a long while and has achieved great successes [16,18,19], although single electroporation such as that after fertilization eggs/embryos were electroporated only once with the plasmid was also feasible. In the absence of TALEN plasmids, the same procedures were utilized to generate the control group. After electroporation and fertilization, the embryos were transferred into 10 L tubs with 7.0 L Holtfreter’s solution containing 10 ppm doxycycline for hatching.

### 2.3. Mutant Detection amd Plasmid Integration Inspection for P_1_ Fish

The pelvic fin and barbel samples were collected for DNA extraction when the fish reached 6 months old [20]. Firstly, the Roche Expand High FidelityPlus PCR System (Roche, Indianapolis, IN, USA) was used to amplify the *fsh* gene utilizing the following primers, reverse primer 5′-CAGAATTCCGTGGCCATTTA-3′; forward primer 5′-CACAACTCCAGCTGTGACAA-3′ (Appendix A). The PCR program was set as follows: 94 °C 2 min; 94 °C 30 s; 60 °C 30 s; 72 °C 40 s; go to step 2 for 29 cycles; and 72 °C 10 min.

The Surveyor^®^ Mutation Detection Kit (Integrated DNA Technologies, Coralville, IA, USA) was utilized to test gene mutation [21,22]. Surveyor Nuclease has proven to be an effective and repeatable tool for mutation testing. It has been used to detect mutations and polymorphisms in the genomes of humans, mammals, bacteria, and plants [23,24]. TA cloning and Sanger sequencing were conducted to confirm the presence of the mutations. The details were described in Qin et al. 2022 [16]. Ten single colonies were randomly selected and cultured in LB broth. After overnight shaking, samples were sent to Eurofins Genomics Company for sequencing. The T-coffee online tool was used to align the DNA sequences.

Integration of plasmids into the fish genome or retention of the plasmids in the cytoplasm was further investigated by PCR; two sets of primers directed to the CMV promoter and TAL repeats region of the plasmids were utilized for amplification (Appendix A). The PCR conditions are as follows: 94 °C 2 min; 94 °C 30 s; 60 °C 30 s; 72 °C 40 s; go to step 2 for 29 cycles; and 72 °C 10 min for final extension.

### 2.4. Fish Culture and Mutant Detection for F_1_ Fish

F_1_ egg masses were put in hatching troughs that had aeration and constant water flow. To achieve a 40–50 ppm hardness, calcium chloride solution was continuously added into the hatching trough using home-made dripping equipment [25]. Two hours after spawning collection, a paddlewheel was used to gently stir the eggs. The egg masses were given a prophylactic treatment every 8 h with 100 ppm formalin or 32 ppm copper sulfate to inhibit the growth of fungus [26]. Twelve hours before the hatch, the treatments were stopped.

With water temperatures ranging from 26 to 28 °C, fish embryos started hatching after one week. They consumed their yolk sac three days after hatching and started the swim-up stage. Before being stocked into a recirculating system, they were given artemia (Brine Shrimp Eggs, Carolina Biological) three times daily.

Pelvic fins were sampled from 30 fish in each family when the 2016 spawned F_1_ fish were 1.5 years old. When 2017 spawned F_1_ fish were 1 year old, ventral fins were sampled from 20–30 fish in each family. Surveyor mutation analysis was conducted following the procedures described in 2.3.

### 2.5. Reproductive Evaluation for P_1_ fish

When the fingerlings were 15–20 cm long, they were moved to a recirculating system or a clay pond to continue growing and maturing. Mutant fish were selected if they had detected mutations and polymorphisms in amplified DNA. On 20 June 2016, mating tests were performed. Five mutant (M) males and three mutant females were selected to mate with wild-type (Wt) females and males (Wt♀ × M♂, M♀ × Wt♂). Five pairs of the Kansas Random strain control channel catfish were chosen for natural spawning without hormone injection. Each pair of fish were kept in a 48 cm × 36 cm × 21 cm overhead aquarium with compressed air and had a steady water flow. Excellent secondary sexual traits were present in all fish. Males that were sexually mature possessed large and reddish genital papillae, well-developed head muscles, and black mottling on their lower jaw and abdomen. Compared with immature females, females at full maturity displayed a broader, rounder belly and a reddening genital region.

In order to test the fertility of mutant males, only the wild-type females were mated with the mutant males in aquariums and implanted with 75 μg/kg LHRHa to induce spawning. Then, 90 μg/kg LHRHa was re-implanted in the wild-type females that did not spawn within 8 days. If the females did not spawn after the first two implants, they were replaced with other wild-type females twice, and the replaced females were implanted for ovulation. To determine the fertility of the mutant females, they were allowed 17 days to mate with males of the wild-type and spawn naturally, without hormone injection.

### 2.6. Hormone Therapy for P_1_ fish

Two *fsh* mutant males and two *fsh* mutant females mated with each other and were implanted with 50 μg/kg LHRHa and 300 IU/kg pregnant mare gonadotropin (M♀ × M♂) in 2016. The additional mating design between mutant female with wild-type male and wild-type female with mutant male was implemented for fertility evaluation. Hormone therapy was carried out in 2017 after it was determined that infertility existed in 2016. Nine mutant males and ten mutant females with exceptional secondary sexual traits were chosen to mate together (M♀ × M♂). Human chorionic gonadotropin (HCG) (1200 IU) was injected into both males and females. Males and females were injected with 400 IU HCG once again if they did not spawn within 5 days. Five pairs of wild-type channel catfish as controls were selected to perform natural spawning without injecting the hormone.

### 2.7. Pleiotropic Effects Evaluation

The survival rates of P_1_ embryos, P_1_ fingerlings, and P_1_ adults were calculated. F_1_ fish that were 1 and 1.5 years old had their survival rate and body weight measured. A low dissolved oxygen incident in one tank resulted in dead fish and 30 of them were sampled. The reproductive behavior of P_1_ fish was observed regularly through the transparent bottom of overhead fish aquariums. Normally, courtship behavior includes the male catfish using their tails to cover the female’s eyes and body quivering.

### 2.8. Statistical Analysis

To calculate and compare body weight, hatch rate, and fish survival rate, the R studio program was used. To assess mutation, survival, and hatch rates when the sample size was small, the Fisher’s exact test was used. The hatch and spawning rates of P_1_ fish were compared between treatments and controls using Student’s *t*-test. Student’s *t*-test was also used to compare body weight in each F_1_ family and survival and mutation rates among F_1_ families before and after hormone therapy. The normality of the data was examined using the Shapiro–Wilk test. A *p* value of 0.05 was the threshold for statistically significant comparisons, and all data were given as the mean ± standard error (SEM).

## 3. Results

### 3.1. Mutation Evaluation and Plasmid Integration in the P_1_ Fish

In 31 of 57 P_1_ fingerlings that were 6 months old, the mutation rate was 54.4%. Five bands were visible on the gel based on the surveyor mutation assay finds. The *fsh* gene was edited at the right position because a band of 350 bp was resolved on the gel (Figure 1 and Appendix A).

The mutation of the *fsh* gene in channel catfish was validated by sequencing and sequence comparison. All the modifications happened within the TALEN cutting region. In the coding area of the *fsh* gene, we found distinct mutation types, including substitutions, base deletions, and insertions, which determines the biological responsiveness of the hormone (Figure 2A). The *fsh* gene’s product is expected to be affected by these mutations, which are also expected to prevent the gene from functioning properly (Figure 2B). We compared these five frame shift mutations with that of the known normal human *fsh* gene structure; four of five resulted in losing of the N-glycosylation sites, which act as the attaching sites for oligosaccharides to bind and then assembly subunit and dimerization afterwards. Therefore, in silico analysis indicates the protein structures might be destroyed and the biological function of *fsh* may be reduced or eliminated (Figure 2C). The correct assembly of the subunit determined the metabolic clearance of the gonadotrophin.

Because neither sequence from the TAL repeats region nor CMV promoter region could be PCR amplified in the plasmid integration assessment, it suggests that no plasmid DNA was present using PCR for all TALEN-*fsh*-defective fish. The results concluded that none of the tested samples harbored the foreign DNA (Figure 3 and Appendix A).

### 3.2. Fertility Evaluation and Hormone Treatment and Intervention

Without hormone intervention, three of five (60.0%) in the control groups reproduced with a mean hatch rate of 74.3% in 2016. Fish receiving LHRHa and pregnant mare gonadotropin did not spawn. In 2017, without LHRHa hormone therapy, five of seven (71.4%) pairs of controls spawned and an average of 32.3% eggs were hatched, whereas using LHRHa hormone treatment, two of three (66.7%) pairs of fish spawned and an average of 56.5% eggs were hatched (Table 1).

Spawnability of P_1_ mutants was evaluated without hormone treatment in 2016. There are 33.3% P_1_ female mutants and 40% P_1_ male mutants spawned, with egg hatch rates ranging from 0.45 to 1.0%. The hatch rate differed considerably between the mutated group and the control group (*p* = 0.001). In 2017, a 1200 IU HCG hormone treatment was implemented. Hormone therapy appeared to improve the spawning rate, 80.0% for female mutants and 88.9% for male mutants, which was similar to that of controls, which were 71.4% without LHRHa injection (*p* = 1.000) and 66.7% with LHRHa injection (*p* = 0.737). A second 400 IU of HCG was injected into the female and male mutants that did not spawn after 5 days; however, the females still did not lay any eggs. F_1_ embryo mean hatch rates for both mutated females and males that underwent hormone treatment were improved to 35.0%, which did not differ from controls, which were 32.3% without LHRHa treatment (*p* = 0.259) and 56.5% with LHRHa treatment (*p* = 0.351) (Table 1).

### 3.3. Mutation Evaluation for the F_1_ Fish

There was one F_1_ family (M♀ × Wt♂) obtained in 2016 and seven F_1_ families (M♀ × M♂) obtained in 2017. The mutations were successfully inherited by the offspring, with inheritance rates ranging from 45.5% to 64% (Table 2). Surveyor assay results showed multiple bands for mutants (Figure 4 and Appendix A).

### 3.4. Pleiotropic Effects

Pleiotropic effects were evaluated based on behavior, hatch rate, earlier fingerling survival rate, later adult survival rate in two environments, and bodyweight. For P_1_ embryos and 6-month-old fingerlings, no substantial differences were found for hatch rate (*p* = 0.760) and survival rate (*p* = 0.602) between the plasmid electroporated group and the control group. For 4-year-old P_1_ fish, mutants did not differ from non-mutants in survival rate cultured in a recirculating system (*p* = 1.000), whereas mutants had higher survival rates than non-mutants when cultured in the pond (*p* = 0.023) (Table 3).

The adult fish with outstanding secondary sexual characteristics were selected for the spawning trial. Abnormal reproductive behaviors were observed in three of the five male mutants and two of the three female mutants. The mutant males and wild-type females were seen engaging in an anomalous courtship between the wild-type females and the mutant males that was visible through the transparent bottoms of aquariums. Females of the wild-type had been observed laying immature eggs without forming an egg mass. In 2017, after receiving 1200 IU of HCG hormone, normal reproductive behavior was seen in two of five (40.0%) mutant males and one of three (33.3%) mutant females.

For the offspring, six families showed no difference regarding body weight between mutant fish and non-mutant fish (Table 2). However, the family 2017 FSH-3 showed a lower body weight for non-mutants (*p* = 0.009). The survival rate was not different among F_1_ families except one family of 2017 *fsh*-5 (*p* = 0.110). A low dissolved oxygen incident happened in one family of 2017 FSH-5 that killed thirty fish in one tank, in which there was detected a 56.7% mutation rate. This mutation rate for the dead progeny did not differ from their siblings (*p* = 0.475) in the family of 2017 FSH-5 in another tank.

## 4. Discussion

At the targeted region of the *fsh* gene, various mutations were identified, such as base deletions, substitutions, and insertions, all of which were efficiently transmitted to F_1_ offspring. Indels have the potential to cause a nonsense codon to inhibit the function of *fsh*. In a scenario, a single nucleotide in *fsh* was altered, which led to a significant impact on the function and expression of the gene because some nucleotide alterations either caused the coding sequence to terminate or mutate into a new codon corresponding a specific amino acid. The spawning rate of P_1_ mutants and their egg hatch rates were suppressed. The P_1_ brood stocks were mostly mosaic. This is also the explanation about how reproduction capacity was not completely eliminated. For the F_1_ generation, genome editing should happen in all the tissues. Hormone therapy with HCG reversed the sterility of P_1_ mutants. Seven families of F_1_ offspring exhibited no evident pleiotropic effects, but only one family of F_1_ offspring showed a decreased body weight for mutants, which might be attributed to the small sample size. Four-year-old P_1_ mutants had a greater survival rate.

TALENs are typically developed to target genes with great precision and optimized to pose a minimal amount of cytotoxicity [27,28,29,30]. In the P_1_ generation, the TALEN-treated group had a similar embryo hatch rate to that of the control group; a comparable fry survival rate was also observed between treated and control groups, indicating that the highly precise TALEN plasmids targeting catfish *fsh* were exceedingly specific and had minimal unintended consequences. The observation that the mortality percentage of mutated P_1_ fish and their full-sibling controls maintained in recirculating systems was the same even after four years provides more support for this conclusion. P_1_ mutants exhibited a higher survival rate in the pond than their full-sibling controls.

The TALENs’ target sites were designed and intended to disable the translation of the β subunit of the *fsh* gene, which provides the biological uniqueness of the hormone. Another possibility is that an aberrant variation in this multi-subunit structure is incapable of working in concert with the remaining subunits, rendering the whole molecule inactive. A mutant of this kind would react biologically as an antimorph, causing disruption to proteolytic activity in a dose-response manner [31,32]. The greatest amounts of *fsh* production and release correlate to the vitellogenesis stage of the salmon’s annual breeding process [12]. The membrane receptors for *fsh* are located on the granulosa and theca cells of the follicle. When *fsh* binds to its receptors on these cells, it regulates the production of hormones (estrogens and androgens), the growth and maturity of follicles, and ultimately the overall functioning of the ovaries. Amago salmon (*Oncorhynchus rhodurus*) [33] and murrel (*Channa punctatus*) [34] are two fish species that have proven establishment of their particular binding sites in the ovary. The ablation of the *fsh* gene should lead to a suppression of the production and release of pituitary gonadotropin, which in turn would cause a reduction in the production of steroid hormones, notably estradiol and testosterone.

Notwithstanding these findings on *fsh* in teleosts over the past 20 years, our understanding of the functions of *fsh* in directing fish reproduction remains limited. In salmonids, it has been hypothesized that *fsh* is primarily accountable for ovarian development, whereas *lh* is essential for triggering final oocyte maturation and ovulation [6,35,36]. Catfishes have a cyclical breeding system with various stages governed by a hormone regulatory signaling pathway, comprising predominantly gonadotropin-releasing hormone (*gnrh*), *lh*, growth hormone, melatonin, and sex steroid hormones. In nature, a gonadotropin burst typically promotes natural oocyte maturity, ovulation, or spermiation [37]. The effects of sex steroids on the teleost pituitary are complex and vary with the species studied, the animal’s physiological status or sex, and dose or mode of delivery [38]. The G protein-coupled receptor that responds to follicle-stimulating hormone is termed the FSH receptor (GPCR). When FSH hormone binds to the receptor on the cell surface, it initiates a chain of events within the cell by activating one or more G proteins, which are signal-transducing proteins that bind to guanine nucleotides. This activation controls the expression of genes, the release of hormones, and the access of ion channels, among other functions [39].

Using TALENs in zebrafish (*Danio rerio*), the hormone-specific β-subunits of both *fsh* and *lh* were individually or collectively knocked out. For *fsh*-deficient zebrafish, the establishment of the ovary, testis, and initiation of puberty was postponed. However, both sexes are still fertile. Also, *fsh* appeared to be involved in retaining the female condition, as sexual reversal was detected in the *fsh*-ablation zebrafish. The dual disruption of the *fsh* and *lh* genes resulted in all male offspring, albeit with severely delayed testicular development [40]. Our results with channel catfish vary from those with zebrafish. This is not entirely unexpected given the diversity of fish reproductive strategies [41]. Some *fsh*-deficient channel catfish with exceptional secondary sexual traits are unable to reproduce due to aberrant reproductive behaviors. The F_1_ embryo from *fsh*-deficient P_1_ females or males had a low hatch rate without HCG hormone therapy, whereas both spawning and hatch rates increased after hormone therapy.

However, whether this is also holds true for other groups of teleosts, such as cyprinids, has yet to be determined, particularly considering that the species-specificity of Fsh receptors varies considerably. In coho salmon (*Oncorhynchus kisutch*), the *fsh* interacted less with the type II receptor (Lh receptor) and associated exclusively with the type I receptor (now known as Fsh receptor) [42]. Purified and recombinant Lh and Fsh were able to increase intracellular cAMP levels in the African catfish (*Clarias gariepinus*) [43]. The scenario is substantially more convoluted in the Indian carp (*Labeo rohita*) because the ovarian follicles’ enriched gonadotropin receptors reacted to both salmon Fsh and Lh but preferred their respective ligands [44].

Glycoprotein hormone receptors (GPHRs) contain three subfamilies, including the luteinizing hormone/choriogonadotrophin receptor (LHCGR), follicle-stimulating hormone receptor (FSHR), and the thyroid-stimulating hormone receptor (TSHR) [45,46]. Equine chorionic gonadotropin, like LH, FSH, and thyroid-stimulating hormone, is a member of the glycoprotein family and its hormone has been employed to promote reproduction in horses and cows [47]. Chorionic gonadotropins are only found in primates and Equidae have both LH-like and FSH-like activity [48,49]. Various fish species are reproductively diverse, and HCG has been used to spawn many fish species, including channel catfish, but with variable success depending upon the species [49,50,51,52]. HCG and a cocktail of Fsh/Lh were demonstrated to be equivalently effective in increasing oocyte maturation and inhibiting the beginning of degeneration in grey mullet (*Mugil cephalus* L.) [53]. In perch (*Perca fluviatilis*), similar situations were observed after hormonal treatment of the common carp, *Cyprinus carpio*, with pituitary extract and HCG [54]. For therapeutic purposes, both Lh and HCG have been employed because of the activities they trigger while binding to the Lh/HCG receptor. This interaction is triggered by the fact that the two Lh/HCG hormones share the same Lh/HCG receptor. Because of its unique structure, the Lh/HCG receptor is known to be a member of the superfamily of G protein-coupled receptors [55]. Interacting with specific surface receptors, Fsh stimulates the development and proliferation of both theca cells and granulosa cells [56].

Although GTHs determine the steroidogenic potency, which in turn regulates FSH production and secretion, sex steroids do not alter the transcription of the *Fshβ* gene in rats in in vitro conditions [57]. In ovariectomized sheep, estradiol therapy represses the expression of *Fshβ* mRNA [58]. In the current research, the bottom line is sterility, which is the ultimate goal. Therefore, the direct effect of an FSH knockout mutation on the HPG axis, the GTH axis, and steroidogenesis may be very species-specific and complicated. Future research will need to examine whether the *fsh* gene knockout has any pleiotropic effects on the levels of *Fsh* mRNA, protein, or hormone in both control and mutant fishes before and after hormone therapy.

## 5. Conclusions

In the current investigation, TALEN plasmids targeting the channel catfish *fsh* gene were introduced into the sperm and early embryos through double electroporation. The mutagenesis of *fsh* was detected at the genomic level and resulted in sterile channel catfish via a TALEN-mediated effect. Using fish mating trials, the reproductive capacity of *fsh*-mutant channel catfish were evaluated and found to be negligible. F_1_ generation fish were subjected to mutation assays to confirm the inheritance of the mutations. No substantial pleiotropic effects were discovered in the F_1_ generation. Correction of sterility caused by the mutated *fsh*-gene in channel catfish is feasible with hormone treatment. The successful development and demonstration of this sterilization technology could be used for the bioconfinement of catfish and other genetically engineered fish, as well as domestic and invasive fish.

## Figures and Tables

**Figure 1 biology-12-00392-f001:**
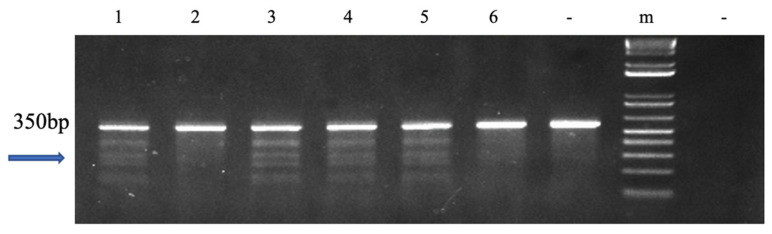
Surveyor mutation detection test results of P_1_ channel catfish (*Ictalurus punctatus*). The mutation was targeted at the follicle-stimulating hormone (*fsh*) gene. The “-” on the left of the molecular marker indicates the control, which is wild-type fish DNA. The right “-” denotes the negative control using water as the DNA template; “m” indicates 1 kb molecular marker. Mutated channel catfish are shown in lanes 1, 3, 4, and 5, and wild-type channel catfish are displayed in lanes 2 and 6. Figure 1 was derived from the full-length gel shown in Appendix A.

**Figure 2 biology-12-00392-f002:**
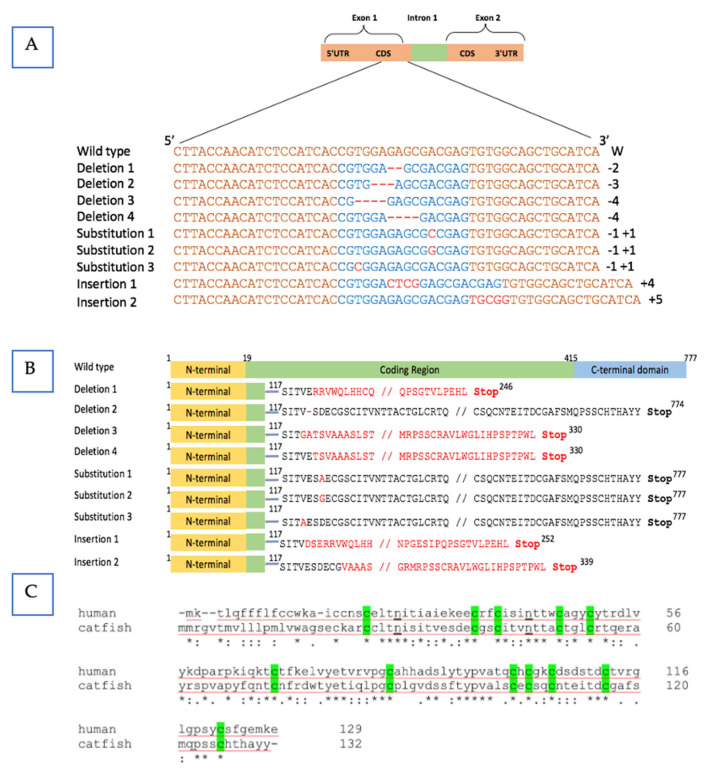
Sequence analysis of the mutated follicle-stimulating hormone (*fsh*) gene in channel catfish (*Ictalurus punctatus*). (**A**) Sequence analysis of the nucleic acid in the mutated *fsh* gene. The channel catfish *fsh* nucleotide sequence of the wild-type is displayed at the top. Orange highlighted sequences represent the target binding sites of the transcription activator-like effector nucleases (TALENs). Blue colored sequences represent the predicted cleavage sites of the nucleases. The mutation types (deletion/insertion/substitution) of nucleotides are shown by red dashes and letters. Numbers of nucleotides deleted (-) or inserted (+) in the edited *fsh* gene are denoted on the right side of the sequences. (**B**) Sequence analysis of corresponding predicted amino acid in the *fsh* mutants. Frameshift reading resulted in a premature stop codon (red colored). Single amino acid deletions or substitutions were caused by changes in one nucleotide mutations or three nucleotide deletions that are highlighted in red. (**C**) Bright-green shaded areas are the Cys residues, and N-linked glycosylation sites are underlined. The asterisk “*” indicated the same amino acid between normal human and wild-type catfish.

**Figure 3 biology-12-00392-f003:**
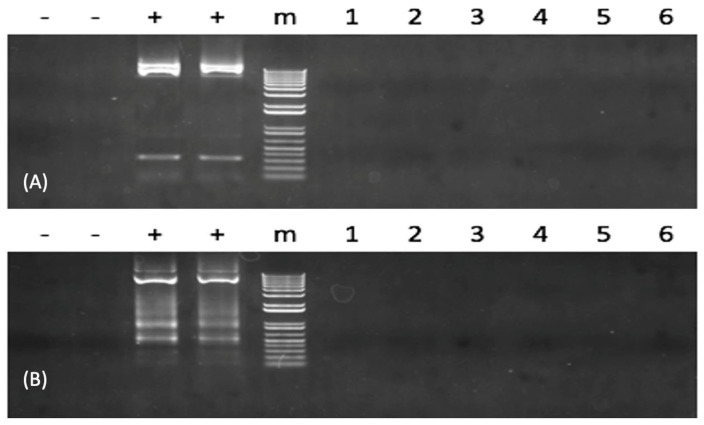
Analysis of transcription activator-like effector nuclease (TALEN) plasmid integration into the genome of channel catfish (*Ictalurus punctatus*) using polymerase chain reaction (PCR) genome. In (**A**,**B**), from the left to the right lanes of “-” denote the negative controls. The left one indicates using water as a template, and the right one indicates the wild-type channel fish sample; the two “+” denote the positives. The left one indicates the TALEN plasmids containing the left DNA binding site, and the right one indicates TALEN plasmids containing the right DNA binding site; “m” indicates 1 kb molecular weight standards. The same six *fsh*-mutated channel catfish individuals (numbered 1–6) were checked for plasmid DNA element integration. The cytomegalovirus (CMV) promoter region was resolved in (**A**). The transcription activator-like (TAL) repeats region was resolved in (**B**). The original full gel image of Figure 3 can be found in Appendix A.

**Figure 4 biology-12-00392-f004:**
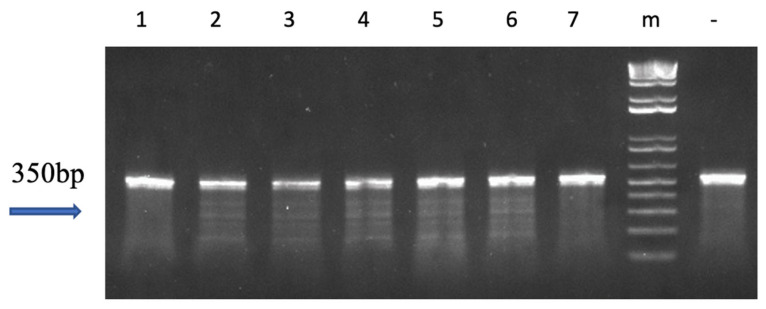
Mutation analysis of follicle-stimulating hormone (*fsh*) gene edited F_1_ channel catfish (*Ictalurus punctatus*) using the Surveyor mutation detection test. “-” represents the wild-type DNA as negative control; “m” represents 1 kb DNA molecular weight standards; numbers (2, 3, 4, 5 and 6) show the mutated channel catfish individuals; lanes numbered 1 and 7 show progeny without anticipated mutation. The full gel picture of Figure 4 can be found in Appendix A.

**Table 1 biology-12-00392-t001:** Spawnability evaluation in terms of spawning rate and embryo hatch rate in two spawning seasons, 2016 and 2017. For the purpose of evaluating spawning, two different types of controls were used: the non-hormone-injected control (nCTRL) and the injected-hormone control (iCTRL) using a 90 μg/kg luteinizing hormone releasing hormone analog (LHRHa) implant. In 2016, wild-type fish were coupled with 3 mutant females and 5 mutant males. The 90 μg/kg LHRHa was only given to the wild-type female fish. In 2017, 10 female mutants were paired with 9 mutant males. A total of 1200 IU human chorionic gonadotropin (HCG) was given to both females and males. To calculate the hatch rate, divide the number of eggs hatched by the number of eggs in total, then multiply by 100. Mean hatch rate data were displayed as the mean of all families ± standard error (SEM).

		Spawning in 2016				Spawning in 2017			
		Fish N	Spawned Fish N	Spawning Rate (%) ^a^	Mean Hatch Rate (%)	Fish N	Spawned Fish N	Spawning Rate ^b^ (%)	Mean Hatch Rate (%) ^c^
*fsh*	F	3	1	33.3	1.0 *	10	8	80.0	35.0 ± 0.14
	M	5	2	40.0	0.5 ± 0.05 *	9	8	88.9	35.0 ± 0.14
iCTRL	M and F	/	/	/	/	3	2	66.7	56.5 ± 0.34
nCTRL	M and F	5	3	60.0	74.3 ± 0.02 *	7	5	71.4	32.3 ± 0.20

* In 2016, there were significant changes in the hatch rate between 2 groups, including the *fsh* gene edited group and the control group (Student’s *t* test, *p* = 0.001). ^a^ In 2016, there were no significant differences regarding the spawning rates among mutant males, mutant females, and wild-type fish (Fisher’s exact test, *p* = 1.000). ^b^ In 2017, there were no significant differences in the spawning rates among mutant fish, iCTRL, and nCTRL after hormone therapy (Fisher’s exact test, *p* = 0.737). ^c^ In 2017, there were no significant differences between the *fsh* mutants and the nCTRL group. There were no significant differences between *fsh* mutants and the iCTRL group after hormone therapy (Student’s *t* test, *p* = 0.259, *p* = 0.351).

**Table 2 biology-12-00392-t002:** Pleiotropic effects evaluation, including survival rate, mutation rate, and mean body weight of channel catfish (*Ictalurus punctatus*). There are a total of 8 families of F_1_ progeny, including one 1.5-year-old 2016 FSH family without performing hormone therapy and seven 1-year-old 2017 FSH families with hormone therapy (2017 FSH-1, FSH-2, FSH-3, FSH-4, FSH-5, FSH-6, and FSH-7). The 2016 FSH family was generated from one *fsh*-mutated female fish paired with wild-type channel catfish without hormone treatment. However, 2017 FSH families were generated from mutant females mated with mutant males with hormone therapy. The 2017 FSH-5 family of fish perished as a result of low dissolved oxygen. Data on body weight are displayed as the mean of the body weight ± standard error (SEM).

Family Names of F_1_ Offspring	Survival Evaluation			Mutation Evaluation			Mean Body Weight (g) Mutant Fish ^a^	Body Weight (g) of Non-Mutant Fish ^a^
	N Fish	N Fish Survived	Survival Rate (%)	N Fish Sampled	N Mutant Fish	Mutation Rate (%)		
2016 FSH	250	222	88.9	30	19	63.3	36.4 ± 2.13	35.6 ± 2.21
2017 FSH-1	22	20	91.0	20	11	55.0	30.2 ± 4.76	28.6 ± 5.26
2017 FSH-2	300	279	93.0	30	15	50.0	9.5 ± 0.55	10.9 ± 0.81
2017 FSH-3	24	20	83.3	25	16	64.0	15.2 ± 1.09 *	22.8 ± 2.19 *
2017 FSH-4	300	266	88.7	21	10	47.6	24.6 ± 3.22	26.6 ± 1.89
2017 FSH-5	300	168	56.0	30	18	60.0 ^b^	12.8 ± 0.82	11.9 ± 0.62
2017 FSH-5 dead fish	/	/	/	30	18	60.0 ^b^	16.67 ± 0.811	17.67 ± 1.509
2017 FSH-6	300	259	86.3	28	13	46.4	18.3 ± 1.48	16.2 ± 0.78
2017 FSH-7	300	269	90.0	22	10	45.5	8.4 ± 0.70	9.3 ± 1.41

* Mutant fish in the family 2017 FSH-3 had a considerably lower body weight (Student’s *t* test, *p* = 0.009) than non-mutant fish in the same family. ^a^ The mutant fish and non-mutant fish had no significant differences in body weight in seven F_1_ families, including 2016 FSH (*p* = 0.807), 2017 FSH-1 (*p* = 0.827), 2017 FSH-2 (*p* = 0.156), 2017 FSH-4 (*p* = 0.585), 2017 FSH-5 (*p* = 0.373), 2017 FSH-6 (*p* = 0.223), and 2017 FSH-7 (*p* = 0.578) (Student’s *t* test). ^b^ Mutation rate was not significantly different (*p* = 0.475) between the 2017 FSH-5 family of fish and the sibling fish that perished as a result of oxygen depletion in this family (Fisher’s exact test).

**Table 3 biology-12-00392-t003:** Pleiotropic effects of the P_1_ embryo, 6-month-old fingerlings and 4-year-old P_1_ fish. Survival rate and hatch rate were calculated. Transcription activator-like effector nuclease (TALEN)-plasmid electroporated and no-plasmid electroporated embryos and fingerlings were compared. Mutant and non-mutant channel catfish (*Ictalurus punctatus*) cultured in different environment were compared.

Treatment	P_1_ Embryos and 6-Month-Old Fingerlings				Genotype	4-Year-Old P_1_ Fish in 2017			
						Recirculating System		Pond	
	N Eggs	N Hatched	Hatch Rate (%) ^a^	Survival Rate (%) ^b^		N Fish	Survival Rate (%) ^c^	N Fish	Survival Rate (%) *
Electroporated with plasmids	200	80	44.0	71.3	*fsh* Mutants	42	54.8	37	54.1
Electroporated without plasmids	200	84	44.0	75.0	*fsh* Non-mutants	2	50.0	6	0

^ab^ No significant differences in hatch rate (*p* = 0.760) and survival rate (*p* = 0.602) were detected between the electroporated with TALEN plasmids group and the electroporated without plasmids group for P_1_ embryos and 6-month-old fingerlings (Fisher’s exact test). ^c^ P_1_ mutants and non-mutants at 4-years of age had similar survival rates (*p* = 1.000) in a circulating system (Fisher’s exact test). However, the sample size is very small for non-mutants. * P_1_ mutants had a higher (*p* = 0.023) survival rate than non-mutants at 4-years of age in the pond (Fisher’s exact test).

## Data Availability

Not applicable.

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
