# Peer review of "Gene Editing of the Follicle-Stimulating Hormone Gene to Sterilize Channel Catfish, Ictalurus punctatus, Using a Modified Transcription Activator-like Effector Nuclease Technology with Electroporation"

_biology, 2023, doi:10.3390/biology12030392_

Round 1

Reviewer 1 Report (Previous Reviewer 1)

The comments are well justified and the text is modified according to the suggestions of the reviewers. For me it is necessary to demonstrate that there is no FSH in plasma to know if the knock out of the gene has worked correctly. But it is true that you have shown that the gene is interrupted and therefore, due to the lack of budget that you have, I accept the article in the present form.

Reviewer 2 Report (Previous Reviewer 2)

I think the topic of this manuscript fits the journal and also the section. The author's response reassures my concerns. They also revised the manuscript to make it more precise. I do not have any questions about the resubmitted manuscript. 

Reviewer 3 Report (Previous Reviewer 3)

The authors have addressed all of my concerns. Therefore, the manuscript is suggested to be accepted.

This manuscript is a resubmission of an earlier submission. The following is a list of the peer review reports and author responses from that submission.

Round 1

Reviewer 1 Report

The manuscript reports the use of modified Transcription Activator-like Effector Nuclease Technology (TALENs) to knock out the Follicle-stimulating hormone (FSH) in Channel catfish, Ictalurus punctatus. The hypothesis is that, with this technology, sterility can be induced in genetically modified fish to limit gene flow into the natural environment. Finally, the results show that mutant fish were sterile and reversal of sterility in gene-edited fish is achievable through hormone therapy. They conclude that this sterilization technology could be used for the bioconfinement of Channel catfish and other genetically engineered fish, as well as domestic and invasive fish.

However, they do not specifically demonstrate knockout of the fsh gene by measuring levels of Fsh mRNA or protein in both control and mutant fishes, and after hormone therapy. They do not measure levels of steroid to see an effect of the mutations in the brain-pituitary-gonad axis and in the reproduction. The number of replicates in the experiment is not enough to conclude, only by visualizing the courtship and with spawning experiments, that sterility is only due to the mutation. There should be different types of controls; 1) with the absence of TALENs plasmids as they do and 2) injecting the TALENS plasmid but without the sites to target β subunit of Channel catfish fsh gene to see an effect of the plasmid itself. Therefore, for me the conclusions are not well supported by the results.

The state of the art and the discussion are not adequate. There is a lack of bibliography on gonadotropins and their role in the reproduction of the Channel catfish. Only with a quick search in databases I have found different works, talking about gonadotropins and their receptors, levels during the reproductive cycle and even the production of recombinant gonadotropins in Channel catfish, which are not mentioned in the entire manuscript.

It is necessary to review the format of the bibliography.

In teleosts the genes are written in lower case and italics, FSH gene is misspelled throughout the manuscript.

For all the above mentioned I will reject the manuscript to be published.

Reviewer 2 Report

The manuscript “Gene Editing of the Follicle-Stimulating Hormone Gene to Sterilize Channel Catfish, Ictalurus punctatus, Using a Modified Transcription Activator-like Effector Nuclease Technology with Electroporation” is very interesting. As a basic scientist, I think the experimental design is fine and the results are properly interpreted. Based on the existing results, I think this manuscript is worth to published after revision if the author could provide more solid information regarding the economic advantages (including but not limited to yield, better disease resistance, greater survival ability in extreme environment).

Concerns:

1.      Technique questions. Are there any advantages over Cas9-based genome editing? Is there any reason for picking sperm and fertilized egg, not oocyte and fertilized egg? To my understanding, sperm genome should be more compact than oocyte genome. Genome editing should be harder in sperm. Why do you use double electroporation technique? Does it work if using single electroporation in fertilized egg?

2.      There are some misinterpretations in the manuscript. Authors should carefully revise the sentences. For example, Line 53-54 in Page 2.

3.      Does genome editing happen in all the tissues or only in some tissues?

4.      “P1 mutants had spawning rates of 33.3% for females and 40% for males when there was no hormone therapy with egg hatch rates of 0.45-1.0%”. Is that suggesting FSH is not 100% required for spawning and hatching?

5.      The inheritance rate is in the range of 45.5%-64%. Not 100%. Is that suggesting that genome editing did not happen in the reproductive system of some mutants? This also raised the question of how many generations the inheritance can insist.

6.      Proof of principle, the strategy provided in this paper is fine. I am wondering if it is economically applicable. Could you provide more information regarding this?

Reviewer 3 Report

In this manuscript, the authors generated FSH mutated channel catfish using TALEN gene editing technology. The manuscript is interesting and well-organized. There are some issues need to be addressed before it is accepted.

(1) Table 2, why the survival rate of non-mutants was 0?

(2) Did the authors check the serum levels of FSH and LH?

(3) For the hormone therapy, HCG and LHRHa mimic the function of LH, the authors should make a discussion why  HCG and LHRHa could the spawning defects of FSH mutant channel catfish.
